# Establishment of Wild-Derived Strains of Japanese Quail (*Coturnix japonica*) in Field and Laboratory Experiments

**DOI:** 10.3390/biology12081080

**Published:** 2023-08-02

**Authors:** Tatsuhiko Goto, Satoshi Konno, Miwa Konno

**Affiliations:** 1Department of Life and Food Sciences, Obihiro University of Agriculture and Veterinary Medicine, Obihiro 080-8555, Hokkaido, Japan; 2Research Center for Global Agromedicine, Obihiro University of Agriculture and Veterinary Medicine, Obihiro 080-8555, Hokkaido, Japan; 3Volunteer Bander, Division of Avian Conservation, Yamashina Institute for Ornithology, Abiko 270-1145, Chiba, Japan

**Keywords:** captive animal, bird banding, breeding, genetic resource, Japanese quail, migration, wild-derived strain

## Abstract

**Simple Summary:**

Domestic quail have been used for egg and meat production and laboratory experiments. Although the wild ancestors of quail live around the Japanese Archipelago, there is a lack of field studies, especially on reproductive sites. Wild-derived strains of several species have been established to study their domestication history and genetic control. Therefore, this study aimed to investigate wild quails and create wild-derived quail strains. Field observations during the breeding season have provided insights into the wild quails’ migration, reproductive, and social behaviors. Morphological traits were measured in 31 quails (2019–2022 population). By comparing them with the 1914 and 1970 populations, the morphological traits of wild quail stocks in Japan might have been relatively stable over the latest ten decades. In 2021–2022, each two captured wild (Wild) males were mated with both the domestic (Dom) females and hybrids of Dom and Wild (W50) females. We established wild-derived strains of quail (W50 and W75), providing opportunities for future conservation use as wild quail stocks and experimental use in basic science.

**Abstract:**

Domestic quail are used as both farm and laboratory animals. As the wild ancestor of quails is “endangered,” field studies are needed to conserve them. If wild-derived strains of quail are established, they will be unique genetic resources for both farm and laboratory animals. The purpose of the present study was to conduct a field study and create wild-derived quail strains using the breeding stocks in Tokachi, Hokkaido, Japan. Field observations from 2019 to 2022 indicate that wild quails migrate and stay at reproductive sites from late April to late October. Our estimations of the approximate ages of the observed and captured quails imply that adult males have intermittent reproductive opportunities from May to August. Morphological traits collected from adult and juvenile quails in the 2019–2022 population were similar to those previously reported for Japan’s 1914 and 1970 populations. Using natural mating of captured wild males and domestic (Dom) females, we established the W50 and W75 strains, which possessed 50% and 75% genetic contributions from the wild stocks. These unique genetic resources can be applied for future conservation and experimental use to understand the domestication history and genetic basis of quantitative traits.

## 1. Introduction

The Japanese quail (*Coturnix japonica*) belongs to the order Galliformes and family Phasianidae, like the chicken [1,2,3]. In general, the Japanese quail was first domesticated in Japan around the 15th century as a companion animal, especially as a songbird [3]. As reviewed by Minvielle [1] and Tsudzuki [3], the intensive production of Japanese quail began in Japan from the 1900s to the 1920s. Around the 2000s, commercial quail production varied depending on the country: primarily egg production in Asia (e.g., China and Japan) and Brazil, and meat production in Europe (e.g., Spain and France) and the United States. Although the quail industry is quantitatively minor in animal production, producing eggs and meat by quail substantially impacts several countries [1].

Domestic quail has been used for egg and meat production in the quail industry and as laboratory animals in basic science [1,2,3]. Genetic studies of quail as experimental animals began in the 1940s [3]. The advantages of quail in research fields include small body size, low consumption, rapid maturation, and high egg production [2,3]. Many morphological, biochemical, morphogenetic, behavioral, and muscular quail mutants exist [3]. The mode of inheritance of most mutations, such as pigmentation (plumage color and eggshell color), was revealed in the 2000s, and several causative genes have been reported in molecular genetics studies [3]. Recently, the whole genome sequence of quail [4,5] has been analyzed using population genomics [6]. Quantitative trait loci (QTLs) for growth, egg, and meat traits have been revealed so far [7,8,9,10]. Accumulating knowledge about the loci underlying economic traits is essential for understanding the complex genetic basis of animal production [11].

While some wild ancestors of domestic animals (e.g., cattle) have already become extinct, the wild ancestors of domestic quails still live in wild environments surrounding the Japanese Archipelago [12,13]. The quail was assigned as a game bird/hunting target in 1918 and has been captive-bred and released into the wild since the early 1970s [14]. Wild quail migrate and breed in Hokkaido, Japan, as summer birds [15]. Quails were commonly observed in riverbeds and meadows until the mid-1980s in Hokkaido but have rarely been observed since the late 1990s [15]. Since the population of wild quail has been declining significantly since the 1990s, the category of quail was moved to “endangered” in 2006 on the Japanese Red List [16], due to a lack of information (Data Deficient). Researchers using captive wild quails will provide knowledge for conserving the precious genetic resources.

Historically, researchers have investigated body-related traits and genetic diversity in captive wild quails in Japan. Kawahara [12] reported on the body traits of wild quail populations captured near the foot of Mt. Fuji in the Shizuoka Prefecture during the fall migratory season (from the middle of October to the end of November) between 1965 and 1970. With several banders, his research team caught 307 Japanese quail birds using ring-shaped traps and mist nets under the condition that the male birds of the domestic strain were used as a decoy. They measured the body weight and lengths of the external shank, internal shank, wing, tail, and bill, and then compared the data with the 1914 population taken by Kuroda. Kawahara [12] concluded that there were no significant differences between the 1965–1970 and 1914 populations. Kimura and Fujii [17] collected bloods and tissues from three wild quail populations in the Shizuoka, Kochi, and Kagoshima Prefectures. They analyzed polymorphisms in 25 proteins controlled by 34 loci and investigated the genetic distances between the three wild populations and several domestic populations. The three wild populations clustered together and were genetically different from the domestic populations. More than 30 years have passed since these reports were published. Therefore, primary data on the present wild quail populations should be collected.

Wild-derived strains have been established to understand the biological mechanisms in animals, e.g., *C. elegans* [18], *Drosophila melanogaster* [19], and *Mus musculus* [20]. The *C. elegans* Natural Diversity Resource (CeNDR) [21], *Drosophila* Genome Nexus (DGN) [22], Diversity Outbred mice [23], and wild-derived heterogeneous stock (WHS) mice [24] provide us with large opportunities to reveal the genetic mechanisms underlying the quantitative traits because of their higher genetic diversity than the domesticated populations. Therefore, we attempted to establish wild-derived quail strains as novel genetic resources. If wild-derived strains of quail are established, the unique genetic resources will provide us with opportunities for future conservation use as wild quail stocks and experimental use in basic science to understand the domestication history of quail and the genetic architecture of quantitative traits. Thus, the objectives of this study were to investigate the morphological traits of wild quails in a field experiment and to create wild-derived strains of quail from the breeding stocks in the Tokachi area of Hokkaido, Japan, in a laboratory experiment.

## 2. Materials and Methods

This study was approved by the Experimental Animal Committee of Obihiro University of Agriculture and Veterinary Medicine (authorization numbers 18–16, 19–32, 20–141, 21–10, 22–38, and 23–30).

### 2.1. Field Study

A field study of wild quails started in the 2018 breeding season in the Tokachi subprefecture in Hokkaido, Japan (Figure 1). A playback experiment was conducted using the crowing calls from male and female quails to investigate whether wild quails existed. The calls used were generally considered mating calls (crowing) of males and rally calls of females [25,26], and several of them were used arbitrarily. Two examples of calls from the digital sound sources are shown in the spectrograms (Figure 2) created using the sound-editing software Raven Lite (version 2.0; Cornell Lab of Ornithology, Ithaca, NY, USA). In the common quail (*Coturnix coturnix*), male vocalizations function to attract females rather than express territoriality, and females emit attraction (rally) calls in response to male crowing [27]. Using a portable speaker (BK-701, Toshiba Lifestyle Products & Services Corporation, Kanagawa, Japan) and the digital sound sources, quail vocalizations were played for approximately one minute at several locations where quails were expected to live. When quails responded by calling back or showing their appearance near the paths, additional playbacks were conducted to determine whether the individuals were potential capture candidates. Playback was performed near the grassland, meadow, or riverbed paths from April to November, mainly from 04:00 to 08:00. The approximate number of males was roughly estimated by collecting the number of crowing over the playback stimuli and their directions. 

Wild quails were captured during the 2019–2022 breeding seasons (Table 1) with permission from the Ministry of the Environment of the Government of Japan. The capturing method primarily uses playback and covering nets, including a clap net. Investigators looked through field glasses (binoculars) and listened to the crowing to identify the target areas of the quails. Visual observations of free-moving quails were performed using binoculars. A clap net attached to a 50 m long wire was placed in the target area. The clap net can be turned over by manually operating the wire 50 m away if a wild quail appears in the capturing range of the net. After capturing a bird, the investigators immediately caught it within the capturing range. 

Bird banding was performed according to the method described in the bird banding manual [28]. Captive wild Japanese quails were observed to record their sex and age. Morphological traits, including body weight (g), tarsus length (mm), total head length (mm), entire culmen (mm), exposed culmen with cere (mm), natural wing length (mm), maximum wing length (mm), and tail length (mm) were measured using an electronic balance, calipers, and a ruler, following a previously described method [29,30]. Blood samples were collected from the wing veins for future DNA analysis. All banded captive quails were photographed and safely released.

The sex and age of the captured Japanese quails and free-moving juvenile and adult quails were determined by visual observation from the plumage and molt, referring to Lyon [31] and the data on the development of juvenile plumage from growing captive individuals from the chick stage (Konno, unpublished data). Adults and juveniles were defined as individuals that fledged in the previous year or earlier and fledged in the capture year, respectively. Wings in the feather generation (2nd+) indicate that all primaries had molted once or more (second generation or older), observed in adults without a molt limit on their outer primaries. Wings at the feather generation (1st) are worn with breached outer primaries, observed in all juveniles and adults with molt limits on their outer primaries because Japanese quails usually do not molt their outer three primaries in the first year of life. Tails at the feather generation (2nd+) had denser barbs and harder structures observed in all adults and some juveniles. Tails at the feather generation stage (1st) have looser barbs and softer structures observed in some juveniles. For free-moving juveniles, the appearance, including body size and plumage features in wing feathers, abdominal feathers, back feathers, and so on, were used to estimate age.

### 2.2. Laboratory Experiment

Domestic quails (Dom) derived from the quail populations used for egg production were kept on a poultry farm at Obihiro University of Agriculture and Veterinary Medicine. The quails were reproduced based on natural mating to grow young birds and breed the strain using a method commonly used in quail experiments and in the industry. Mixed feed for layers (Rankeeper; Marubeni Nisshin Feed Co., Ltd., Tokyo, Japan) was provided to the adult quails.

For the 2021–2022 breeding seasons, a maximum of two individuals in each season were reared for a while with permission from the Ministry of the Environment of the Government of Japan. Captive wild quail males (two males in 2021 [Wild_01 and Wild_02] and two males in 2022 [Wild_03 and Wild_04]) were reared for a short period (< a month) to mate with females of domestic quail (Dom) and hybrid quails between Wild and Dom (W50), respectively. Natural mating was carried out in the home-cage of a female and a male at a frequency of more than twice per week. After observing the mating behavior, the wild male returned to his home-cage. The fertilized eggs were incubated for 18 days in an artificial incubator (P-03; Showa Furanki, Saitama, Japan). The quails were hatched and reared on ad libitum feed (CP 17%, ME > 2850 Kcal/kg) and water in battery brooders (GS24; Showa Furanki, Saitama, Japan). All the wild-caught males (*n* = 4) were photographed and then released safely.

Pedigree information for the wild-derived quail strains was recorded. We defined the strain names of Dom, W50, and W75, depending on their genetic backgrounds. Dom and W50 had 100% and 50% genetic contributions, respectively, from domestic quail. The W75 strain was produced via backcrossing W50 females with captive wild males. Therefore, approximately 75% of the genetic background of the W75 strain is derived from the wild population.

### 2.3. Statistical Analysis

The basic statistics were calculated using R version 4.2.2 [32] and RStudio version 2022.7.2.576 [33].

## 3. Results

### 3.1. Field Study

A field study was conducted from 2018 to 2022 in Tokachi subprefecture, Hokkaido, Japan. The number of captured wild quails is shown in Table 1. The data were collected from 20 adults (a female and 19 males) and 11 juveniles (five females and six males) from wild quail stocks at the breeding sites in Hokkaido, Japan. Pictures of the captured wild quails (females and males of each age group) are shown in Figure 3. An incubation patch was observed in adult females captured in July 2020 (Figure 3a).

The dates of capture of the wild quails are shown in Figure 4. The number of quails captured in June (11 quails) and August (8 quails) was larger than the others in 2019–2022 in Tokachi subprefecture, Hokkaido. The earliest was in late April, and the latest was in late October. We estimated the approximate ages of the juvenile quails from 2021 to 2022. Age-dependent changes in the molt (feather generation) of the primaries and body size traits have been observed in domestic quails in the laboratory. Visual observations were performed for captured quails in detail, and for uncaptured quails, visual observations were performed using binoculars. From the captured quails, we estimated the approximate ages as approximately 25 days of age (late July), 17 days of age (middle August), and 95 days of age (late October). The ages of the quails were estimated to be around 20 days of age (late June), 20 days of age (late July), 20 days of age (middle August), and one day of age (early September) from the visual observation using binoculars. A nest containing five eggs was also observed in early July (Yonekawa, Personal communication). Male-male interactions (e.g., crowing to each other, walking around the path near the meadows, and dust-bathing together near sand pools) were sometimes observed during the visual observations using binoculars.

### 3.2. Morphological Analysis of Captive Wild Quails 

Body size traits were summarized by groups based on age (adult and juvenile) and sex (female and male) in the captured wild quails (Table 2). As several traits could not be collected, the number of animals (*n*) in the body size traits is shown in the range (Table 2). The body weight of an adult female (118.7 g) tended to be higher than adult males (98.0 ± 5.6 g). The adult quail’s remaining body size traits (tarsus, head, and beak) tended to be comparable between females and males. In juveniles, body size traits tended to be comparable among the sex groups.

Wing and tail traits were summarized by the groups based on the feather generation (2nd+ and 1st) and sex (female and male) in captured wild quails (Table 3 and Table 4). The natural wing length of a female in the 2nd+ groups (92.0 mm) tended to be shorter than that of males (95.3 ± 2.3 mm). In tail length in the 1st group, females (36.5 ± 1.0 mm) somewhat tended to be longer than males (30.2 ± 2.7 mm). The remaining wing and tail length traits tended to be comparable between the females and males.

To check the trend of changing body size in wild quail populations from 1914 to 2022, we included previously reported data from the 1914 and 1970 populations from Kawahara [12]. Since the five body size traits (body weight, internal shank length, bill length, wing length, and tail length) were available from the 1914 and 1970 populations, our five traits (body weight, tarsus length, exposed culmen with cere, natural wing length, and tail length) from the mature individuals (adult or 2nd+ feather age) are shown in the 2019–2022 population (Table 5). Based on the approximate comparison, no remarkable change was observed between the 1914 and 1970 populations and the present populations. 

### 3.3. Establishment of Wild-derived Strains of Japanese Quail

Figure 5 shows that wild-derived strains (W50 and W75) were established in this study. In the 2021 breeding season, two captive adult males (Wild_01 and Wild_02) were mated with five females of the Dom strain to establish the W50 strain. During the 2022 breeding season, two captive adult males (Wild_03 and Wild_04) were mated with four females of the W50 strain to produce the W75 strain. All wild males were kept in a laboratory environment for the mating experiments for a short period (< a month) and then safely released into the wild. As a result, we established novel genetic resources, W50 and W75 quail strains, which possessed 50% and 75% of the genetic background derived from the wild stocks, respectively.

## 4. Discussion

The present study revealed several insights into the migration, reproduction, and social behaviors of endangered wild quails living around the Japanese Archipelago. The combined field and laboratory experiments have enabled the establishment of novel wild-derived quail strains. Accumulating field observations of wild stocks and using novel genetic resources in quantitative genomics studies will provide many opportunities to conserve the wild ancestors and to further understand the domestication history and genetic basis of quantitative traits.

The capture dates ranged from late April to late October. The most frequently captured months were June (11 quails) and August (8 quails) during the breeding seasons in 2019–2022 in the Tokachi subprefecture in Hokkaido, Japan. These captive results may indicate that the wild quails in the breeding season sometimes approach the playback of crowing, which implies that crowing and rally calls may be one of the keys to gathering each other in the vast expanse of riverbeds and meadows. If the quails do not react to the playback, it is impossible to catch the quails in the capturing range of the covering net (approximately 8.0 m × 2.6 m) in the vast field. Since the results of our captive experiment were biased toward males (19 adult males) rather than females (one adult female) in adult quails, the timing of the capture may depend on the levels of social interaction behavior and (or) sexual behavior in adult male quails. The biased results would be influenced by the capture method. However, the speculation of behaviors from our captive experiments was only the tip of the iceberg because the behaviors of wild quails in the meadows often cannot be observed by visual observations (the behaviors can sometimes be observed around the path near the meadows). Therefore, it is necessary to keep collecting field data.

In the present field study, we observed wild quails around the meadows from late April to late October, corresponding to the range of dates captured. These observations will provide insights into the migratory behaviors of Japanese quails. Wild quails seem to arrive and stay at the reproductive sites in Tokachi, Hokkaido, at least in late April and late October. Our estimates of the approximate ages of the wild quails were approximately 20 days of age in late June, 20 and 25 days of age in late July, 17 and 20 days of age in the middle of August, 1 day of age in early September, and 95 days of age in late October. Therefore, the estimated hatching dates will be around early June, early July, middle and late July, and early September. Since the incubation period of quail is 17–18 days [34], the estimated periods for incubation behavior will be around late May–early June, late June–early July, early July–the middle of July, the middle of July–late July, and late August–early September. Lukanov and Pavlova [13] summarized that wild Japanese quails lay about 5–14 eggs per clutch, with 2–3 broods per year. Given that female quails lay around ten fertilized eggs in each clutch, the estimated periods for mating and collecting eggs in nests (nesting behavior) will be around the middle of May–late May, the middle of June–late June, late June–early July, early July–the middle of July, and the middle of August–late August. Since the newborn chicks require warm environments provided by the mother quail, brooding behavior will continue for several weeks from hatching to offspring growth [13]. After the offspring grow, the adult females will move to the next clutch. However, ground-nesting birds living in meadows potentially face several risks of nest loss. When pasture grasses are cut by operators, quail nests, including fertilized eggs, will sometimes be destroyed. After resetting the clutch, females will start to mate with males in the next clutch. These estimates and speculations indicate that the reproductive opportunities for adult male quails continue intermittently from May to August in Tokachi.

Although only one adult female quail was captured between 2019 and 2022, a brood patch (incubation patch) was observed in early July 2020. A nest containing five eggs was observed once in early July 2021. In addition, several juvenile quails were seen from June to October. These observations are good evidence for the reproduction of wild quails in the Tokachi area. Moreover, morphological traits include body size and wing and tail lengths from the 2019 to 2022 wild quail population. Unfortunately, we could not conduct statistical comparisons because the data from the adult females were limited, and the developmental stages of the juvenile quails varied. Since the population size of the wild stocks in Tokachi would be small, field experiments should be conducted in the future. Kawahara [12] compared the morphological traits of the wild quails captured between the 1970 and 1914 populations (1914 data referred from Kuroda). The 1970 population during the fall migratory season near Mount Fuji [12] probably included adult and juvenile quails because the population would just arrive there from the breeding sites. Similar morphological features were observed in an approximate comparison with the wild quail populations from 1914, 1970, and 2019–2022 in Japan. From the results, the morphological traits of the wild quail stocks around the Japanese Archipelago might have been relatively stable over the last ten decades.

When domestic animals and their wild ancestral populations exist, the wild population is a significant genetic resource. A comparison between the domestic and wild populations will provide much knowledge, e.g., what genetic and behavioral changes occurred during the process of domestication in the domestic population. In addition, because the genetic diversity of domestic quail may be relatively low compared to the wild stocks, creating new, highly diverse wild-derived strains would be useful for maintaining and improving the genetic diversity of future domestic quail populations. As long as the wild males can be captured, the genetic diversity of our wild-derived strains will be increased via backcrossing W50 and W75 females with captive wild males. Although the W50 and W75 strains now receive a strong selective pressure to adapt to the laboratory environment to maintain the stocks generation by generation, we observed that the present W50 and W75 strains are well-adapted to the laboratory environments. However, the wild stocks are endangered. Therefore, the wild-derived quail strains, which probably have the genetic makeup of migration behavior and adaptive traits to wild environments, will provide opportunities for future conservation use as wild quail stocks in the laboratory. It is important to continue field studies and capture wild quails to monitor the population size and genetic diversity using DNA analyses to conserve such valuable wild resources for future generations [35].

## 5. Conclusions

Wild-derived populations, including the *C. elegans* Natural Diversity Resource [21], *Drosophila* Genome Nexus [22], Diversity Outbred mice [23], and wild-derived heterogeneous stock mice [24], provide new insights into the genetic architecture underlying the quantitative traits. The Japanese quail (*Coturnix japonica*) is a well-known model organism for studying photoperiodism [36], reproductive physiology and behavior [37], and genetics [1,2,3,35]. Our original wild-derived quail strains may possess a wild-derived genetic makeup of migratory, photoperiodic, and reproductive behaviors, as well as adaptation, activity, and production traits. Future population genomic studies using novel wild-derived quail genetic resources will provide opportunities to understand the genetic mechanisms underlying the migratory behavior, voluntary activity, and egg and meat production.

## Figures and Tables

**Figure 1 biology-12-01080-f001:**
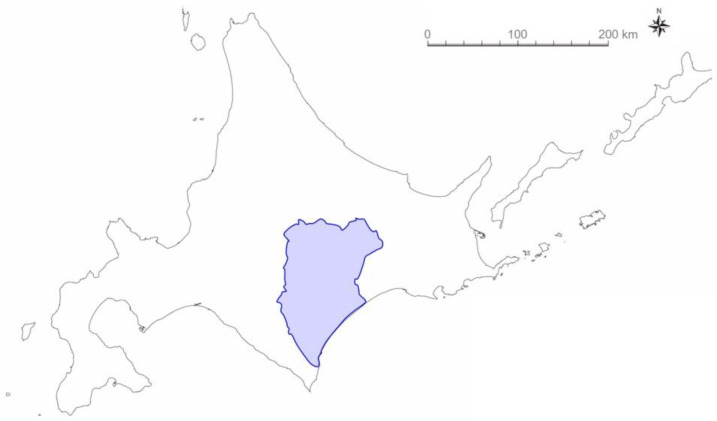
Location of field study for banding wild Japanese quail (*Coturnix japonica*). The Tokachi area (blue) in Hokkaido, Japan, a breeding site for quails. The figure is created from a map (Geospatial Information Authority of Japan).

**Figure 2 biology-12-01080-f002:**
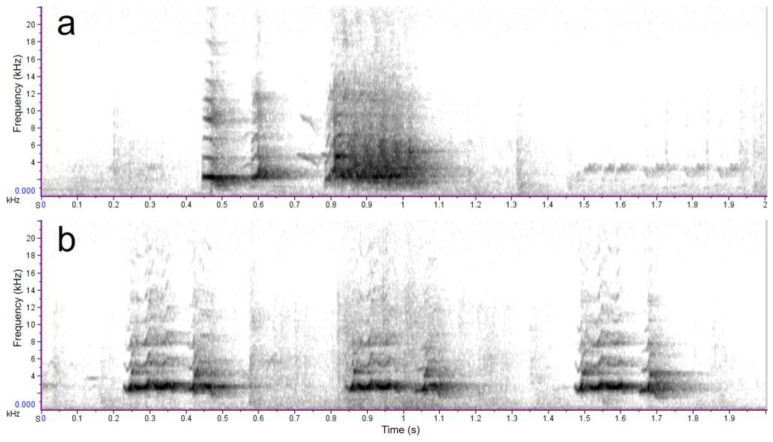
Spectrograms of Japanese quails. (**a**) A mating calls (crowing) for males, and (**b**) a female call (rally call).

**Figure 3 biology-12-01080-f003:**
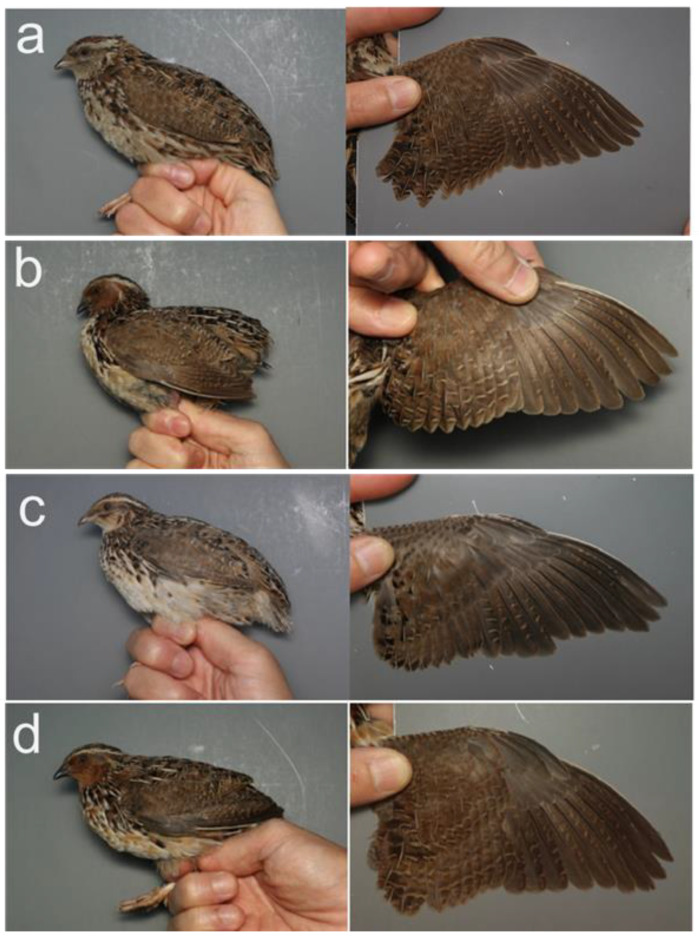
Pictures of captured wild quails. (**a**) Adult female, 3Y+, 20200706, Tokachi, Hokkaido. The throat is grayish brown: originally pale brown but dust-stained. All remiges are in the same generation; no molt limit exists in the outer primaries. (**b**) Adult male, 2Y, 20190618, Tokachi, Hokkaido. The throat is rufous. The first-generation (= juvenile) feathers are retained at the outer three primaries, and the other remiges are the second-generation feathers molted in the previous year. (**c**) Juvenile female, 1Y, 20200819, Tokachi, Hokkaido. The throat is pale brown. The inner three primaries and one secondary are the second-generation (= post-juvenile) feathers, and the other remiges are the first-generation feathers. (**d**) Juvenile male, 1Y, 20190818, Tokachi, Hokkaido. The throat is rufous. The inner four primaries are the second-generation feathers, and the other remiges are the first-generation feathers. Eight digit numbers indicate the date captured (yyyymmdd). The 3Y+, 2Y, and 1Y (ages) mean the individuals fledged in the year before the previous year or more, the previous year, and the same year captured, respectively.

**Figure 4 biology-12-01080-f004:**
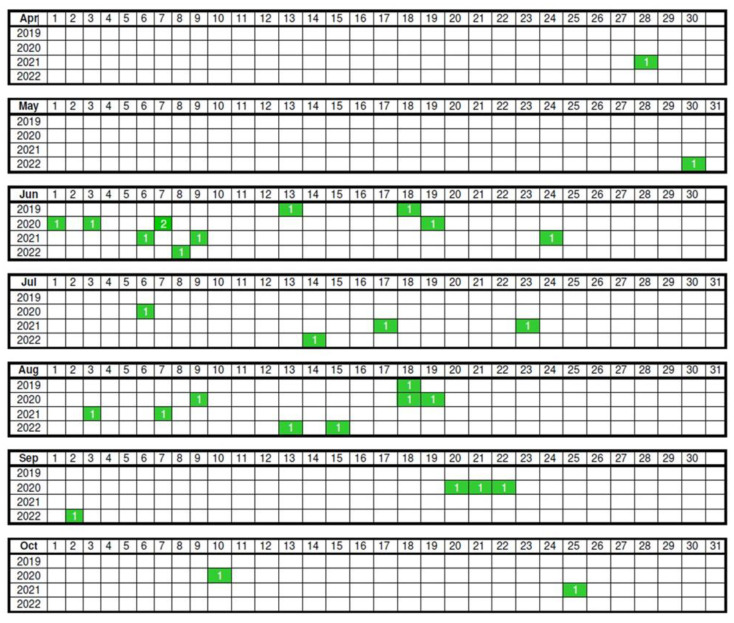
The dates of the wild quails captured. Green squares indicate the dates of wild quail captured each year. The number of animals is shown in the green square. June (11 quails) and August (8 quails) are larger numbers captured. The earliest is late April, whereas the latest is late October.

**Figure 5 biology-12-01080-f005:**
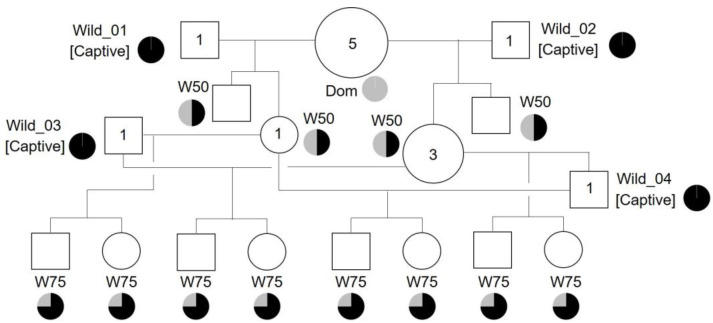
Pedigree chart of wild-derived strains of quail. White square and white circle are male and female, respectively. Numbers in squares and circles are the number of animals used for mating. Strain names are Dom, W50, and W75. The Wild_01, Wild_02, Wild_03, and Wild_04 (captive males) mate with females of Dom and W50, and then are released to the wild. Pie charts indicate a ratio of genetic background derived from wild (black) and domestic (grey) quails.

**Table 1 biology-12-01080-t001:** Number of wild quails captured in Tokachi area, Hokkaido, Japan, from 2019 to 2022.

Year	Number of Birds Captured					
	Adult			Juvenile		
	Female	Male	Total	Female	Male	Total
2019	0	2	2	0	1	1
2020	1	5	6	2	5	7
2021	0	8	8	2	0	2
2022	0	4	4	1	0	1
Total	1	19	20	5	6	11

The sex and age of captured Japanese quails were determined from plumage and molt. Adult and juvenile were defined as individuals which fledged in the previous year or earlier and fledged in the capturing year, respectively.

**Table 2 biology-12-01080-t002:** Measurements of body size traits in captured wild quails from 2019 to 2022.

Trait	Age (Adult)		Age (Juvenile)	
	Female	Male	Female	Male
	*n* = 1	*n* = 16–19	*n* = 4–5	*n* = 6
Body weight (g)	118.7	98.0 ± 5.6	84.3 ± 14.0	90.9 ± 7.0
Tarsus length (mm)	27.0	26.7 ± 1.0	27.0 ± 0.5	27.4 ± 1.1
Total head length (mm)	36.3	35.7 ± 0.6	35.0 ± 0.9	35.9 ± 0.7
Entire culmen (mm)	16.5	15.9 ± 0.7	15.4 ± 0.6	15.4 ± 1.0
Exposed culmen with cere (mm)	13.7	13.5 ± 0.5	12.8 ± 1.0	13.2 ± 0.9

Data are shown as mean ± SD. Individual data are shown in adult females because of one individual. The number of individuals is shown as a range when missing data are included. Body weight is measured in adult males (*n* = 16), whereas entire culmen is collected in juvenile females (*n* = 4). The remaining traits are measured in adult males (*n* = 19) and juvenile females (*n* = 5). Adult and juvenile are defined as individuals fledged in the previous year or earlier and fledged in the capturing year, respectively.

**Table 3 biology-12-01080-t003:** Measurements of wing length traits in captured wild quails from 2019 to 2022.

	Feather Generation (2nd+)		Feather Generation (1st)	
Trait	Female	Male	Female	Male
	*n* = 1	*n* = 5	*n* = 3	*n* = 20
Natural wing length (mm)	92.0	95.3 ± 2.3	95.0 ± 1.9	93.8 ± 1.7
Maximum wing length (mm)	103.5	104.3 ± 1.5	103.8 ± 1.6	102.8 ± 2.8

Data are shown as mean ± SD. Individual data are shown in 2nd+ female because of one individual. Wings at feather generation (2nd+) indicate that all primaries had molted once or more (second generation or older), which is observed in adults without molt limit on their outer primaries. Wings at feather generation (1st) are worn with breached outer primaries observed in all juveniles and adults with molt limit on their outer primaries because the Japanese quails normally do not molt their outer three primaries in the first year of life.

**Table 4 biology-12-01080-t004:** Measurements of tail length trait in captured wild quails from 2019 to 2022.

Trait	Feather Generation (2nd+)		Feather Generation (1st)	
	Female	Male	Female	Male
	*n* = 2	*n* = 18	*n* = 3	*n* = 4
Tail length (mm)	38.0 ± 0.0	37.7 ± 2.5	36.5 ± 1.0	30.2 ± 2.7

Data are shown as mean ± SD. Tails at feather generation (2nd+) are denser barbs and harder structures observed in all adults and some juveniles. Tails at feather generation (1st) are looser barbs and softer structures observed in some juveniles.

**Table 5 biology-12-01080-t005:** Comparison of trait measurements of captured wild quails in Japan from 1914 to 2022.

Trait ^1^	Mean ± SE	Range	Mean ± SE	Range
**1914 population ^2^**				
	Female (*n* = 6)		Male (*n* = 18)	
(1) Body weight (g)	-		-	
(2) Internal shank length (mm)	26.6 ± 0.08	24.1–30.0	26.9 ± 0.03	25.4–30.0
(3) Bill length (mm)	13.1 ± 0.01	13.0–13.5	13.0 ± 0.01	12.7–13.5
(4) Wing length (mm)	98.3 ± 0.04	97.0–99.6	97.1 ± 0.04	94.2–99.6
(5) Tail length (mm)	38.6 ± 0.06	36.7–40.6	38.2 ± 0.05	35.3–41.9
**1970 population ^3^**				
	Female (*n* = 22)		Male (*n* = 17)	
(1) Body weight (g)	99.7 ± 1.51	90.0–115.0	96.4 ± 2.12	86.0–114.0
(2) Internal shank length (mm)	26.6 ± 0.30	23.75–28.90	25.4 ± 0.32	23.40–27.95
(3) Bill length (mm)	12.7 ± 0.11	11.25–13.55	12.6 ± 0.09	12.00–13.35
(4) Wing length (mm)	103.0 ± 0.44	99.80–107.60	102.1 ± 0.56	97.00–107.00
(5) Tail length (mm)	39.7 ± 0.69	33.00–42.85	40.3 ± 0.47	34.95–43.40
**2019–2022 population ^4^**				
	Female (*n* = 1)		Male (*n* = 19)	
(1) Body weight (g)	118.7		98.0 ± 1.3	85.9–106.8
(2) Tarsus length (mm)	27.0		26.7 ± 0.2	24.5–29.1
(3) Exposed culmen with cere (mm)	13.7		13.5 ± 0.1	12.5–14.6
	Female (*n* = 1) ^5^		Male (*n* = 5) ^5^	
(4) Natural wing length (mm)	92.0		95.3 ± 2.3	91.5–97.3
	Female (*n* = 2) ^6^		Male (*n* = 18) ^6^	
(5) Tail length (mm)	38.0 ± 0.0	38.0–38.0	37.7 ± 2.5	31.9–42.3

^1^ (1) body weight, (2) tarsus length, (3) exposed culmen with cere, (4) natural wing length, and (5) tail length. Trait names in 1914 and 1970 populations are referred from Kawahara [12]. ^2,3^ Data from Kawahara [12]. ^4^ Data from this study. Raw data are shown when data are from one individual. ^5^ Data from wings at feather generation (2nd+). ^6^ Data from tails at feather generation (2nd+).

## Data Availability

Data are available upon request to the authors.

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
