# Peer review of "Establishment of Wild-Derived Strains of Japanese Quail (Coturnix japonica) in Field and Laboratory Experiments"

_biology, 2023, doi:10.3390/biology12081080_

Round 1
Reviewer 1 Report
The present study was conducted to survey wild quail that arrived in Tokachi, Hokkaido, Japan, between 2019 and 2020 and to establish new two wild-derived strains, which possess 50% and 75% genetic contributions from these wild quail. The authors performed morphological measurements on wild quail in field study and compared them with records obtained in 1914 and 1979 and considered the morphological traits of wild quail in Japan to be relatively stable over the last 100 years.
The topic of the study is very interesting, and the experimental design is adequate, but some methodological and interpretation concerns need to be addressed before the study can be recommended for publication.
Major issues:
1) There is insufficient detail in the methods of playback experiments. How were vocal playback stimuli prepared and how were they played back? In addition, it is generally understood that female quail do not emit crows, but what kind of sound is the “female crowing” used as playback stimuli in this study? The authors should explain it by showing sonograms. Furthermore, the number of male quails estimated from this experiment is not shown in the results. The number of calls back to the playback of male crowing and/or female vocalization, respectively and the number and sex of quail towards the sound source should be stated.
2) The methods for estimating the age of juvenile birds should be detailed in the methods section.
3) Dr. Kawahara (1978) measured the morphology traits of wild quail that migrated from the north to the grasslands and pastures at the foot of Mount Fuji, from the middle of October through the end of November 1970. Thus, these quail were in the fall migratory season not the breeding season. The authors should clearly state this and interpret what is meant by the lack of change in morphological characteristics between quail in the non-breeding season captured in 1970 and those in the breeding season captured from 2019 to 2022.
4) Page 10, line 270: “These results indicated that the wild quails in breeding season sometimes approach playback of crow, ……. The sentence is unclear. The authors should describe what “these results” indicate.
More specific points:
-Page 1, line 35: (Wild) should be removed.
-Page 2, line 46: songbird should be “song bird”
- The authors should show orientation marks or north marks in Figure 1.
-Page 5, line 167: Please provide more details on the diet. At a minimum, what is the crude protein content?
-Figure 2: The legend of Figure 2 is difficult to understand. It should be stated in such a way that the reader can understand what the abbreviations and dates represent just by reading the legend.
-Page 10 lines 278-280: This should be moved to the Results section.
Author Response
Comments and Suggestions for Authors
The present study was conducted to survey wild quail that arrived in Tokachi, Hokkaido, Japan, between 2019 and 2020 and to establish new two wild-derived strains, which possess 50% and 75% genetic contributions from these wild quail. The authors performed morphological measurements on wild quail in field study and compared them with records obtained in 1914 and 1979 and considered the morphological traits of wild quail in Japan to be relatively stable over the last 100 years.
The topic of the study is very interesting, and the experimental design is adequate, but some methodological and interpretation concerns need to be addressed before the study can be recommended for publication.
We appreciate your contribution to improve our manuscript.
Major issues:
1) There is insufficient detail in the methods of playback experiments. How were vocal playback stimuli prepared and how were they played back? In addition, it is generally understood that female quail do not emit crows, but what kind of sound is the “female crowing” used as playback stimuli in this study? The authors should explain it by showing sonograms. Furthermore, the number of male quails estimated from this experiment is not shown in the results. The number of calls back to the playback of male crowing and/or female vocalization, respectively and the number and sex of quail towards the sound source should be stated.
Response 1:
We added the detailed information about the playback (L114-126). Spectrograms of male and female calls were shown in Figure 2a and 2b, respectively.
We could not estimate the accurate number of male and female quails in the playback experiments because the males were not always reacted to the playback sounds. Females almost do not react. When females appeared near the paths for a while, we have an opportunity to observe the appearance from the far place but often cannot identify whether the quail was male or female. Therefore, we cannot indicate the number of individuals estimated in this paper.
2) The methods for estimating the age of juvenile birds should be detailed in the methods section.
Response 2:
We observed appearance of the wild quails using field grass. Body size and plumage features in wing feathers, abdominal feathers, back feathers, etc were points to estimate the age.
We mentioned as below.
“The sex and age of captured Japanese quails and free-moving juvenile and adult quails by visual observation were determined from plumage and molt to refer to Lyon [29] and data from growing captive individuals from the chick stage (Konno, un-published data).” (L154-157).
“For free-moving juveniles, the appearance including body size and plumage features in wing feathers, abdominal feathers, back feathers, etc. were points to estimate the age.” (L166-168).
3) Dr. Kawahara (1978) measured the morphology traits of wild quail that migrated from the north to the grasslands and pastures at the foot of Mount Fuji, from the middle of October through the end of November 1970. Thus, these quail were in the fall migratory season not the breeding season. The authors should clearly state this and interpret what is meant by the lack of change in morphological characteristics between quail in the non-breeding season captured in 1970 and those in the breeding season captured from 2019 to 2022.
Response 3:
We have already mentioned “captured near the foot of Mount Fuji at Shizuoka Prefecture, during fall migratory season (from the middle of October to the end of November) in 1965-1970.” (L78-79). We think that population in fall migratory season includes adult and juvenile quails because the population will just arrive there from the breeding sites. If possible, we would like to know how the body weights are changed before and after the migration event. We added the discussion (L340-342).
4) Page 10, line 270: “These results indicated that the wild quails in breeding season sometimes approach playback of crow, ……. The sentence is unclear. The authors should describe what “these results” indicate.
Response 4:
We changed “these results” to “these captive results” (L291). We would like to refer the results mentioned in the prior sentence.
More specific points:
-Page 1, line 35: (Wild) should be removed.
Response 5:
We deleted it (L35).
-Page 2, line 46: songbird should be “song bird”
Response 6:
We changed (L46).
- The authors should show orientation marks or north marks in Figure 1.
Response 7:
We added it (Figure 1).
-Page 5, line 167: Please provide more details on the diet. At a minimum, what is the crude protein content?
Response 8:
We added the information (CP 17%, ME > 2850 Kcal/kg) (L185).
-Figure 2: The legend of Figure 2 is difficult to understand. It should be stated in such a way that the reader can understand what the abbreviations and dates represent just by reading the legend.
Response 9:
We added the information as below.
“Eight digits numbers indicate the date captured (yyyymmdd). The 3Y+, 2Y, and 1Y (ages) mean the individuals fledged in the year before the previous year or more, the previous year, and the same year captured, respectively.” (Figure 3).
-Page 10 lines 278-280: This should be moved to the Results section.
Response 10:
We moved the sentence in Results (L230-232). And then, the modified sentence was shown (L299-302).
Submission Date
21 June 2023
Date of this review
28 Jun 2023 14:27:27

Reviewer 2 Report
This study combined field observations and a laboratory experiment aimed at documenting the status of Japanese quails in Japan and the possibility of establishing a wild-derived strain. Given the precarious status of this species in Japan, this is a worthwhile endeavour. Some clarifications are needed and concerns are listed below. The text will benefit from an English revision.
Line 61: Please rephrase this sentence. It is not clear what you mean by ‘has revealed in a population level’.
Line 69: To be clear, are there still natural populations of the quail that are not harvested? Or is it the case that all wild birds are captive-bred? These captive-bred birds might be quite different from the wild stock as there could be selection for traits favourable to captivity and hunting.
Line 125: I think field glasses are known as binoculars.
Line 168: You probable mean the wild-caught males were released, yes?
Line 182+: This section just repeats what is available in Table 1. This is redundant information. Please use one or the other. Any reason why adult females are so rare in the samples? Perhaps this is something to do with the capture method. Alternatives might be discussed as it would be interesting to start a wild-derived strain with females also.
Table 2. It might be preferable to show sample size next to each measurement rather than as a range.
Line 229: It is hard to say that a sex difference occurs given that there was only 1 adult female.
Table 5: At least for the males, there is enough data to make a formal statistical comparison among populations using a linear model with year as a factor. This would be better than just a verbal argument.
Line 263: The lab experiments tell us that it is possible to cross wild males with domestic females. However, how many such crosses would be needed to really start a novel wild-derived strain? For instance, a wild-derived strain derived from four males might suffer from lack of genetic diversity. In addition, it is not clear whether the wild-derived strains adapt well to captive conditions. Please discuss.
Line 271: Careful here as crow is the name of a bird species. You probable mean crowing males.
Needs extensive revision.
Author Response
Comments and Suggestions for Authors
This study combined field observations and a laboratory experiment aimed at documenting the status of Japanese quails in Japan and the possibility of establishing a wild-derived strain. Given the precarious status of this species in Japan, this is a worthwhile endeavour. Some clarifications are needed and concerns are listed below. The text will benefit from an English revision.
We appreciate your contribution to improve our manuscript.
Line 61: Please rephrase this sentence. It is not clear what you mean by ‘has revealed in a population level’.
Response 1:
Papers [4,5] analyzed whole genome sequence from one individual, but the paper [6] analyzed 31 individuals from three populations (wild, egg-type and meat-type). Therefore, we wanted to show them. We modified the sentence as below.
“Recently, the whole genome sequence of quail [4,5] has analyzed by population genomics [6].” (L61-62).
Line 69: To be clear, are there still natural populations of the quail that are not harvested? Or is it the case that all wild birds are captive-bred? These captive-bred birds might be quite different from the wild stock as there could be selection for traits favourable to captivity and hunting.
Response 2:
We think that the 31 wild quails captured in this study are wild stocks (not captive-bred quails released). The quail was out of game birds in 2007 (Japan Hunters Association). Okuyama (2004) mentioned the releasing captive-bred quails have stopped in 1991 (Tokyo) and 1997 (Chiba). Therefore, the present wild quail stocks would maintain their populations by natural mating, brooding, and migration over 25 years.
Actually, we cannot rule out the possibilities of the introgression of the Dom genetic background into the wild quails in the past. In addition, we think that the W50 and W75 strains now receive the strong selective pressure to adapt to the laboratory environments for maintaining the stocks generation by generation.
Line 125: I think field glasses are known as binoculars.
Response 3:
We added it (L140).
Line 168: You probable mean the wild-caught males were released, yes?
Response 4:
Thank you. We modified the sentence as ”All the wild-caught males (n = 4) reared were taken photos and then were released safely.” (L185-186).
Line 182+: This section just repeats what is available in Table 1. This is redundant information. Please use one or the other. Any reason why adult females are so rare in the samples? Perhaps this is something to do with the capture method. Alternatives might be discussed as it would be interesting to start a wild-derived strain with females also.
Response 5:
We deleted three sentences (L200).
Although the wild males probably are active to the playback of crowing, the wild females may not active than male. We discussed about it (L294-304). If possible, we will try to capture adult females for creating the new wild-derived strains. But, we expect that it should be very difficult to collect fertilized eggs because we do not know whether the wild adult females produce eggs after mating under the laboratory environments.
The females were less likely to respond to the attraction, which is influenced by the capturing method. Since quail is a rare species, we considered it undesirable to capture females, especially during breeding (possibly during egg-laying or brood-rearing). In addition, the number of quails that can be temporarily reared is limited to two individuals per year, and the males were captured early. In the future, we would like to capture juvenile females and attempt to use them to develop a strain derived from wild females.
We added the discussion (L297).
Table 2. It might be preferable to show sample size next to each measurement rather than as a range.
Response 6:
We collected body weight in adult male (n = 16) and entire culmen in juvenile female (n = 4). The other traits were collected in adult male (n = 19) and juvenile female (n = 5). Therefore, we described the information in footnote (Table 2).
“Body weight is measured in adult male (n = 16), whereas entire culmen is collected in juvenile female (n = 4). The remaining traits are measured in adult male (n = 19) and juvenile female (n = 5).”
Line 229: It is hard to say that a sex difference occurs given that there was only 1 adult female.
Response 7:
We think so. But, the data is precious for us. Therefore, we just would like to indicate tendency (not statistical difference) using the terms “tended to be higher” and “tended to be shorter” in body weight and natural wing length, respectively.
We discussed that “Unfortunately, we could not conduct statistical comparisons because data from adult females were limited, and the developing stages of the juvenile quails were varied.” (L334-336).
Table 5: At least for the males, there is enough data to make a formal statistical comparison among populations using a linear model with year as a factor. This would be better than just a verbal argument.
Response 8:
We gave up to conduct a linear model with year as a factor because individual data cannot be available from 1914 and 1970 populations.
Line 263: The lab experiments tell us that it is possible to cross wild males with domestic females. However, how many such crosses would be needed to really start a novel wild-derived strain? For instance, a wild-derived strain derived from four males might suffer from lack of genetic diversity. In addition, it is not clear whether the wild-derived strains adapt well to captive conditions. Please discuss.
Response 9:
Under the endangered status of the wild quails, we think that the W50 strain using two wild males is precious resource. Fortunately, we succeed the backcrossing using two wild males to make the W75 strain (75% wild genetic background). This is a different meaning from that Dom strain females mated with four wild males. Generally speaking, the wild ancestor population has much larger genetic diversity than domesticated population. As long as the wild males can be captured, the genetic diversity of the wild-derived strains will be increased by backcrossing the W50 and W75 females into the captive wild males. From now, we will analyze genomic information of the stocks to monitor the diversity.
Regarding the adaptation to captive condition, we observed the W50 and W75 strains look well-adapted to the lab environments. But, we think that the W50 and W75 strains now receive the strong selective pressure to adapt to the laboratory environments for maintaining the stocks generation by generation.
We added the discussion (L352-357).
Line 271: Careful here as crow is the name of a bird species. You probable mean crowing males.
Response 10:
We changed (L292).
Comments on the Quality of English Language
Needs extensive revision.
Response 11:
We re-received by grammar editing.
Submission Date
21 June 2023
Date of this review
23 Jun 2023 15:04:19

Round 2
Reviewer 1 Report
In their revised version of the manuscript by Goto et al., the authors could handle several weak points. Still, the description of their methods and the interpretation of their results is not always precise.
- In the method l. 114, the sonagram of female vocalizations used as playback stimuli is very helpful in understanding the experimental methodology. Based on the sonogram and the previous studies presented below, we consider that the female vocalization used as playback stimuli is either "rally calls" or "distress calls".
Authors should cite the following papers instead of those of closely related species (25).
Dere´gnaucourt and Guyomarc’h / Ethology. 2003 109:107–119
Chiba and Hosokawa / Hormones and Behavior. 2006 49: 4–14
Desmedt et al., Ethology. 2020 126:553–562.
- In the method l. 156, data from growing captive individuals from the chick stage (Konno, unpublished data) is unclear. Is it data on morphology? As it is unpublished data, the reader cannot easily refer to it, so it should be indicated what the data are about.
- In the discussion l. 289 they state: “Range of the dates captured were from late April to late October. The more frequent months captured were June (11 quails) and August (8 quails) during breeding seasons in 2019-2022 at Tokachi subprefecture in Hokkaido, Japan. These captive results indicated that the wild quails in breeding season sometimes approach playback of crowing, which implying the crowing is one of the keys to gather each other in the vast expanse of riverbeds and meadows.”
It is difficult for me to understand the logic of how the results showing the period wild quail were captured and the month when they were captured the most suggest that playback of the crowing of male quail elicits phonotaxis in quail.
Furthermore, the authors were unable to indicate the number of quail that responded to the playback of quail calls in the result section. In addition, in the Authors_response, they state: "the males were not always reacted to the playback sounds. Females almost do not react." Therefore the results are not reliable to make such conclusions.
Another reviewer strongly recommended English editing and the author stated that he complied, but the revised manuscript did not confirm any improvement in the quality of the English language.
Author Response
We have modified the manuscript. We would be happy if you re-evaluate our manuscript.

Reviewer 2 Report
Thank you for taking my comments into account. I have no further comments.
While the revised version has improved, extensive edition is still needed.
Author Response
Thank you very much for your contribution to improving our manuscript.
Round 3
Reviewer 1 Report
The authors addressed almost all my comments and now the manuscript is much improved.